# What Is the Impact of Mass Timber Utilization on Climate and Forests?

Rachel Pasternack [1,*], Mark Wishnie [2], Caitlin Clarke [1], Yangyang Wang [1], Ethan Belair [1], Steve Marshall [3], Hongmei Gu [4], Prakash Nepal [4], Franz Dolezal [5], Guy Lomax [6], Craig Johnston [7], Gabriel Felmer [8], Rodrigo Morales-Vera [9], Maureen Puettmann [10] and Robyn Van den Huevel [11]

[1]  The Nature Conservancy, 4245 North Fairfax Drive, Suite 100, Arlington, VA 22203, USA; caitlin.clarke@tnc.org (C.C.); yangyang.wang@tnc.org (Y.W.); ethan.belair@tnc.org (E.B.)
[2]  BTG Pactual Timberland Investment Group, 1180 Peachtree Street NE Suite #1810, Atlanta, GA 30309, USA; mark.wishnie@btgpactual.com
[3]  Mass Timber Strategy, LLC, Washington, DC 20016, USA; steve@masstimberstrategy.com
[4]  USDA Forest Service Forest Products Laboratory, One Gifford Pinchot Drive, Madison, WI 53726, USA; hongmei.gu@usda.gov (H.G.); prakash.nepal@usda.gov (P.N.)
[5]  IBO—Austrian Institute for Building and Ecology, 1090 Vienna, Austria; franz.dolezal@ibo.at
[6]  Global Systems Institute, College of Engineering, Mathematics and Physical Sciences, University of Exeter, Exeter EX4 4PY, UK; guylomax01@gmail.com
[7]  Independent Researcher, Ottawa, ON K2G 0V1, Canada; craigmtjohnston@gmail.com
[8]  Instituto de la Vivienda, Facultad de Arquitectura y Urbanismo, Universidad de Chile, Santiago 8331051, Chile; gfelmer@uchile.cl
[9]  Centro de Biotecnología de Los Recursos Naturales (CENBIO), Facultad de Ciencias Agrarias y Forestales, Universidad Católica del Maule, Talca 3480112, Chile; rmorales@ucm.cl
[10]  CORRIM—Consortium for Research on Renewable Industrial Materials, Corvallis, OR 97339, USA; maureen@corrim.org
[11]  Dalberg Catalyst, Johannesburg 2196, South Africa; robyn.vandenheuvel@dalberg.com
*  Correspondence: rachel.pasternack@tnc.org

**Abstract:** As the need to address climate change grows more urgent, policymakers, businesses, and others are seeking innovative approaches to remove carbon dioxide emissions from the atmosphere and decarbonize hard-to-abate sectors. Forests can play a role in reducing atmospheric carbon. However, there is disagreement over whether forests are most effective in reducing carbon emissions when left alone versus managed for sustainable harvesting and wood product production. Cross-laminated timber is at the forefront of the mass timber movement, which is enabling designers, engineers, and other stakeholders to build taller wood buildings. Several recent studies have shown that substituting mass timber for steel and concrete in mid-rise buildings can reduce the emissions associated with manufacturing, transporting, and installing building materials by 13%-26.5%. However, the prospect of increased utilization of wood products as a climate solution also raises questions about the impact of increased demand for wood on forest carbon stocks, on forest condition, and on the provision of the many other critical social and environmental benefits that healthy forests can provide. A holistic assessment of the total climate impact of forest product demand across product substitution, carbon storage in materials, current and future forest carbon stock, and forest area and condition is challenging, but it is important to understand the impact of increased mass timber utilization on forests and climate, and therefore also on which safeguards might be necessary to ensure positive outcomes. To thus assess the potential impacts, both positive and negative, of greater mass timber utilization on forests ecosystems and emissions associated with the built environment, The Nature Conservancy (TNC) initiated a global mass timber impact assessment (GMTIA), a five-part, highly collaborative research program focused on understanding the potential benefits and risks of increased demand for mass timber products on forests and identifying appropriate safeguards to ensure positive outcomes.

**Keywords:** mass timber; carbon storage; lifecycle analysis; regional demand assessments; global trade modelling; climate change; forest impact assessments; sustainable forest management

## 1. Introduction

As the need to address climate change grows more urgent, policymakers, businesses, and others are seeking innovative approaches remove carbon dioxide emissions from the atmosphere and decarbonize hard-to-abate sectors. Concrete and steel, construction materials whose combined production represents about 11 percent of annual global greenhouse gas emissions, present a particular challenge [1]. Global building stock, which is primarily composed of these materials, is projected to double over the next 40 years, effectively adding a built area the size of Paris to the planet every week through 2060 [2]. Aligning this projected surge in construction with the climate mitigation goals of the Paris Agreement is critical to a climate-stable future. Forests can play a role in reducing atmospheric carbon. However, there is disagreement over whether forests are most effective reducing carbon emissions when left alone versus managed for sustainable harvesting and wood product production.

Timber framing and "post-and-beam" construction are traditional methods of constructing buildings. Historically, this type of construction has been limited to low-rise buildings such as single-family homes, smaller apartment buildings, and non-residential structures. More recently, there has been a growing interest in building more with wood. A new class of wood products (mass timber) has emerged, allowing wood buildings to be much taller (e.g., 8–18 stories), and thus mass timber has the potential to displace some steel and concrete building materials, which today have inherently higher embodied carbon and energy. Cross-laminated timber (CLT) is at the forefront of the mass timber movement, which is enabling designers, engineers, and other stakeholders to build taller wood buildings. CLT panels are made by laminating dimension lumber orthogonally in alternating layers. Panels generally made from CLT are lightweight, yet very strong, with good fire, seismic, and thermal performance [3,4].

Several recent studies have shown that substituting mass timber for steel and concrete in mid-rise buildings can reduce the emissions associated with manufacturing, transporting, and installing building materials by 13–26.5% [5–7]. Other studies have quantified the amount of carbon stored in mass timber materials themselves, which persists for the useful life of the building and perhaps longer if materials are recovered, reused or repurposed [8].

However, the prospect of increased utilization of wood products as a climate solution also raises questions about the impact of increased demand for wood on forest carbon stocks, on forest condition, and on the provision of the many other critical social and environmental benefits that healthy forests can provide. Increased wood harvest for mass timber use can increase, decrease, or have a neutral impact on forest carbon stock, depending on the forest attributes and environmental factors, the harvest and management strategies, the spatial and temporal scale being viewed, the carbon pools being considered in the forest ecosystem, and indirect impacts on the wider wood product market [9,10]. For example, increased demand for forest products through sustainable harvesting may expand forest carbon sinks by encouraging forest growth and regeneration over time [11,12]. It can incentivize new tree planting and investment in forest management that can contribute to increased forest growth and inventory [13]. Improved forest management may lower the risk of wildland fires in regions such as the western U.S. [14–16], which are increasing in intensity potentially reducing overall forest carbon stocks and threatening forests and communities. However, increased demand may also have negative impacts, if for example, unsustainable forest management is adopted, by altering harvest intensities or rotation length beyond sustainable levels. Increasing mass timber demand can potentially also have initial negative impacts on forest carbon stocks through increased production emissions and residues.

A holistic assessment of the total climate impact of forest product demand across product substitution, carbon storage in materials, current and future forest carbon stock, and forest area and condition is challenging. Several recent studies have tried to assess the total climate impact of changes in wood demand across the full value chain at regional or national levels, concluding that improved forest management and shifts to longer-lived

wood product utilization would drive net climate benefits in Canada [17–20] and in selected sites across North America [21,22]. Other researchers have concluded that utilization of long-lived wood products could drive net negative impacts on climate, excluding product substitution benefits [23,24].

For policymakers, developers, and others considering the use of mass timber to achieve climate and policy goals, this lack of clarity can be confusing. Additionally, use of mass timber is generally projected to increase due to general market forces. For these reasons, it is important to understand the impact of increased mass timber utilization on forests and climate, and therefore also on which safeguards might be necessary to ensure positive outcomes.

## 2. Global Mass Timber Impact Assessment (GMTIA)

To assess the potential impacts, both positive and negative, of greater mass timber utilization on forests ecosystems and emissions associated with the built environment, The Nature Conservancy (TNC) initiated a global mass timber impact assessment (GMTIA), a five-part, highly collaborative research program focused on understanding the potential benefits and risks of increased demand for mass timber products on forests and identifying appropriate safeguards to ensure positive outcomes.

We selected five regions with high potential for mass timber utilization based on a range of criteria: we selected two regions in which mass timber has already achieved modest levels of adoption (Europe, which represented 60% of the global mass timber market in 2018 [25], and the USA, where 576 mass timber projects have either been built or are currently under construction [26]. We chose additional regions where recent actions suggest that mass timber may play a role in future climate policies [27–29]; including one region in which global construction activity is projected to be concentrated through 2030 (China, which represents 24% of total projected global floor area expansion through 2016–2030 [2]); and one region that is home to significant areas of commercial softwood forests [30] and a well-established forest products manufacturing industry, but where mass timber markets remain nascent (the southern cone of South America)..

The GMTIA is organized in five phases of work (Figure 1):

i.   Comparative life cycle assessments (LCAs) of functionally equivalent mass timber and conventional buildings in selected regions (Europe, China, Chile, and the US) to estimate embodied carbon and carbon storage of mass timber utilization at the individual building level for representative buildings using designs that are locally appropriate to each region, but functionally equivalent across regions. As with most LCAs, phase 1 of the GMTIA does not consider impacts on forest carbon stocks, which are explicitly addressed in phase 4. These LCAs also do not consider end-of-life treatment.

ii.  Regional demand assessments to extend the results of individual building LCAs to estimate embodied carbon, carbon storage, and changes in wood demand at varying levels of mass timber adoption (conservative, optimistic, or extremely high adoption levels) in new construction in each of the selected regions.

iii. Global trade modelling using a variant of the Global Forest Products Model to estimate how changes in demand for forest products associated with increased penetration of mass timber in each region will directly and indirectly impact global forest product trade flows (e.g., if 90% of new buildings in region X are built with mass timber, where will that timber be supplied from, and will other trade flows be displaced?).

iv.  Forest impact assessments to evaluate the spatial-temporal impact of mass timber harvests on forest composition, structure and carbon stocks in forest ecosystems associated with different predicted mass timber demand scenarios as indicated by the regional demand assessment and the global trade modelling in Phases ii and iii.

v.   Integration of the results of Phases 1–4 to estimate the total impact on climate and forests of different levels of mass timber utilization in the selected regions and

the identification of potential risks and of conditions needed to reduce potential negative impacts. Results of all phases will also be communicated via academic articles and policy recommendations.

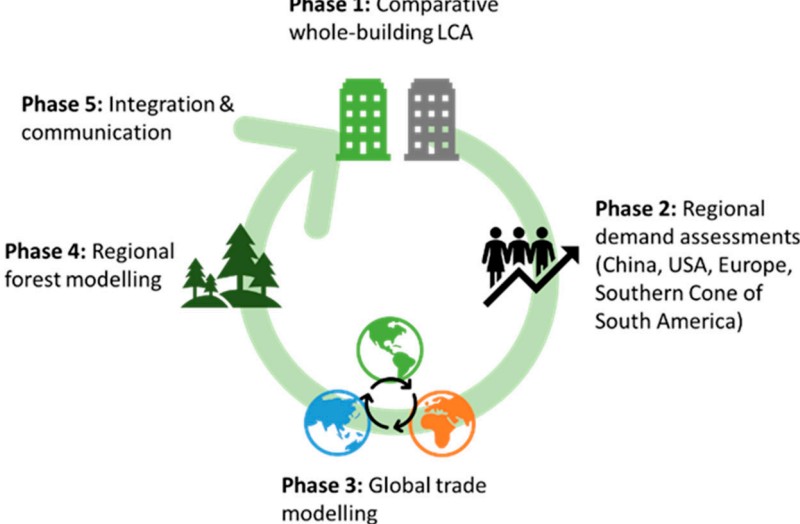

**Figure 1.** Five phases of the global mass timber impact assessment (GMTIA).

In July 2018, The Nature Conservancy convened a collaborative multi-disciplinary group of forest ecologists, conservation practitioners, academics, economists, and lifecycle analysts to design a comprehensive approach to understand the total impact of greater mass timber utilization on forests and climate. We convened over 20 collaborators and partners, bringing in a wide array of knowledge and expertise on the complex issues that need to be considered to assess the impacts of increased demand for mass timber (see the acknowledgements section for a full list of collaborators). The remainder of this article briefly discusses the theoretical basis for this research, which occurs in five phases. The initial three phases of this research make up many papers within this Special Issue of *Sustainability*.

### 3. Theoretical Basis

The climate impacts of concrete and steel are typically calculated as the emissions associated with the extraction, processing, manufacturing, transportation, installation, use, maintenance, and disposal of the products (often referred to as embodied emissions) for which traditional LCA is well suited [31,32]. However, because of all the ecosystem services forests can provide, understanding the full climate impacts of forest product utilization requires consideration of a wider range of factors, not all of which are captured in typical LCA methodologies:

a **Forest carbon stock changes**: Changes in demand for mass timber may drive market level changes that increase or decrease forest carbon stocks, forest area, or both, as described above. Demand changes may also simply drive product shifts or changes in utilization rates, and thereby have no detectable impact on forest carbon stocks. These impacts are likely to vary based on geography, the forest composition and structure, existing forest management practices, magnitude of demand changes, forest tenure, forest plans and ownership, timescale, and a variety of other factors.

b **Forest health and climate change**: Demand changes will also lead to changes in forest health (potentially positive or negative). A changing climate will also affect the health of the forest, especially as natural disturbances such as wildfire, insects, and disease increase with intensity. The implications of climate change on forest dynamics ultimately impact the amount of forest products that are produced.

c   **Embodied carbon:** Embodied carbon for the construction products refers to all greenhouse gas (GHG) emissions associated with extraction, processing, and manufacturing, transporting, and installing construction materials [33]. The harvest, transportation, and production of mass timber may have higher or lower emissions than alternative construction materials, thereby generating a negative or positive climate impact, depending upon the process and energy mix of their manufacture, transportation distances, and the emissions associated with the materials for which they are a substitute.

d   **Carbon storage in wood products:** Carbon storage occurs differently based on the type of harvested wood product. Forest products may store carbon for extended periods of time (as in the case of wooden furniture, and mass timber in buildings), emit carbon immediately (as in the case of bioenergy), or store carbon for an intermediate period dependent on recycling (for example, short-term storage in paper products). Temporary carbon storage, however, is not accounted for in traditional LCA frameworks, which tend to treat emissions as equivalent regardless of when in the life cycle they occur.

e   **End-of-Life (EoL):** There are many possible waste scenarios for building materials all of which vary depending on material, local, state, and country regulations, and existing deconstruction standards. Wood products have the potential to be (1) re-used (e.g., wooden boards salvaged from one building for use in another), (2) substituted for energy (e.g., wooden boards salvaged from a building and processed/burned to produce bioenergy), or (3) landfilled. Materials that are fully reused can have climate benefits, while wood products that decay in a landfill can decay slowly, and may produce methane, a very potent greenhouse gas [34].

Estimating the total impact of increased wood product utilization on climate change, and the potential impacts on forests requires an understanding of abovementioned factors, and the complex interactions with one another. Little and differing information is currently available to inform decision makers regarding (i) the potential scale of climate impacts associated with greater forest products utilization, (ii) the risks associated with increased wood products demand on forest degradation or deforestation, (iii) the potential benefits of increased wood products demand on reforestation or other increases in forest carbon stock, (iv) factors or conditions that enhance positive impacts or reduce negative impacts on forests and climate, (v) the role of market mechanisms in mediating forest impacts; and (vi) measures that might be taken to maximize benefits, minimize risks, and safeguard against undesirable outcomes [35].

While the GMTIA attempts a comprehensive LCA assessment comparing manufacturing emissions among functional equivalent buildings, it does not consider the operational emissions (the emissions associated with operating and maintaining the building over its useful life) of different building types due to a lack of readily available data and tools to achieve the comparison [32]. Operational emissions represent as much as 28% of the global energy-related $CO_2$ emissions (the main source of emissions in whole building LCA studies) [36], as such we recommend the development of tools and data collection to assess potential operational differences.

## 4. Discussion and Conclusions

Partial results of the first three phases of the GMTIA appear in this Special Issue of *Sustainability*. This Special Issue includes studies that present the results of the comparable LCAs for functionally equivalent buildings of different heights and from different regions; estimates from four important international wood producing regions of the impacts of wood product demand when moderate to high levels of mass timber adoption are considered and estimates of the impacts of these demand changes on global prices, production, consumption, and trade of forest products.

Phases 4 and 5 of the GMTIA, which will generate the results perhaps most critical to decision makers and society at large, necessarily build on the results of the first three phases presented here. Phase 4, the impact assessment of demand changes on forest composition,

structure, and carbon stocks, and phase 5, the integration of phases 1–4 to estimate the total impact of mass timber demand changes on forests and climate and identify pathways to mitigate negative consequences for forests, people, and climate, are already underway. These results are expected in early 2023.

This work represents an important first step toward understanding the full breadth of impacts that increased mass timber utilization could have on forests and climate mitigation around the globe. The series of projects detailed in this Special Issue have been collaboratively designed to answer pressing questions in the discussion of mass timber impacts, both negative and positive. That said, the science, marketing, and application of mass timber are actively expanding fields, and the GMTIA is not able to provide the final word on these topics. Rather, we hope to initiate the development of key research and safeguarding standards that can guide policymakers, commercial interests, and land managers and owners in making sound decisions in how they approach elements of the broader mass timber conversation. It is critical to our success that the methods and standards we develop continue to evolve in response to the best available science, policy, and case studies. This project forms part of a broader portfolio of work and has been developed with a wide range of academic institutions, Non-Government Organizations (NGOS), and consultancies. To further collaborative engagement, The Nature Conservancy would like to hear from additional voices on these issues and encourages those with an interest in collaborating in this on-going work to contact the corresponding author.

**Author Contributions:** Conceptualization, R.P., M.W., C.C., Y.W., E.B., M.P., G.L. and S.M.; Formal Analysis: Y.W., E.B., P.N., F.D., G.L., C.J., G.F., R.M.-V., M.P. and H.G.; Writing—review and editing and project administration: R.P., M.W., Y.W., E.B. and R.V.d.H. All authors have read and agreed to the published version of the manuscript.

**Funding:** This research was funded by USDA Forest Service, Southern Region, Wood Innovations Program (19-DG-11083150-023).

**Acknowledgments:** This article makes up part of a larger 5-phased project which was initiated by The Nature Conservancy (nature.org) through generous support from The Climate and Land Use Alliance, and the Doris Duke Charitable Foundation (DDCF). The work upon which this project is based was also funded in whole or in part through a grant from the USDA Forest Service, Southern Region, Wood Innovations Program (19-DG-11083150-023). The Nature Conservancy initiated this project to further the collective understanding of the potential benefits and risks of increasing demand for forest products and ensuring that any increases are sustainable. The Conservancy's objectives are focused on delivering critical safeguard frameworks to mitigate any potential risks on forest ecosystems as mass timber demand increases. Building designs and assumptions were developed by atelierjones llc with contributions from the USDA Forest Products Laboratory and Woodworks. The building LCAs were completed by teams at the University of Washington (PNW), USDA Forest Products Laboratory (NE), and the Consortium for Research on Renewable Industrial Materials (CORRIM) (SE), with additional support and contributions from Coldstream Consulting. The LCAs developed outside of the US were developed by: University of Washington, IBO—Austrian Institute for Building and Ecology, Universidad de Chile. and Universidad Católica del Maule. University of Washington developed the regional demand assessments. The USDA Forest Products Laboratory, in conjunction with Craig Johnston led the global trade modelling. Phase 4 is being led by The Nature Conservancy. Phase 5 is being led by the Michigan State University, Department of Forestry, Forest Carbon & Climate Program.

**Conflicts of Interest:** The authors declare no conflict of interest. The funders had no role in the design of the study; in the collection, analyses, or interpretation of data; in the writing of the manuscript, or in the decision to publish the results.

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
