# Peer review of "What Is the Impact of Mass Timber Utilization on Climate and Forests?"

_sustainability, doi:10.3390/su14020758_

Round 1
Reviewer 1 Report
This review is good and aligned with the current topic of climate change.
I only have a few feedback for the authors:
Lines 60 - 72: These short paragraphs can be combined into 1 paragraph.
Line 68: add abbreviation (CLT) after Cross-laminated timber
Lines 82-896: These sentences need literature support.
Line 119: For transparency, I suggest authors inform who are these collaborating organizations.
Line 120: The explanation of 'regions with high potential for mass timber utilization' is general. I suggest the authors provide facts or statistics that support the selection of China, Europe, Chile and the US as the chosen focus region.
Line 129: GMIAT should be GMTIA
Line 146-147: Please specify the 'several important forest regions".
Explain why forest impact assessment initially needs to be done in the US, why not initially do it in Chile etc.?
Lines 195 - 203: Check the line and paragraph spacing
Line 218: Is it a must to capitalize each word of 'Embodied Carbon' in this sentence? In other sentences, it is written in lower case.
Line 225: Need to remove extra spacing between 'sustainability.' and 'This..'.
Author Response
Please see in the attachment.

Reviewer 2 Report
Based on the premise that it is a review article, which proposes to discuss theoretical and methodological assumptions about the use of wood in larger constructions and these impacts on the economy, markets, ecology, environment and climate have little improved and developed , few authors in the text. I believe that the article, as it is a relevant topic, needs to be complemented, expanded with more results, etc. Or, the Information here could be included as an introduction or part of another article by researchers on the subject, considering that of the 5 steps of the method, only 3 are completed.

Round 2
Reviewer 2 Report
When evaluating the changes in the article, I consider that the changes have improved the version.